# The Relationship between Physical Fitness and Perceived Well-Being, Motivation, and Enjoyment in Chinese Adolescents during Physical Education: A Preliminary Cross-Sectional Study

**DOI:** 10.3390/children10010111

**Published:** 2023-01-05

**Authors:** Wei Zheng, Hejun Shen, Mohammed B. Belhaidas, Yaping Zhao, Lei Wang, Jin Yan

**Affiliations:** 1School of Physical Education, Guangzhou Sports University, Guangzhou 510500, China; 2School of Physical Education and Humanity, Nanjing Sport Institute, Nanjing 210014, China; 3Institute of Physical Education and Sports, University of Abdelhamid Ibn Badis, Mostaganem 27000, Algeria; 4The Library Unit, Shandong Sport University, Jinan 250102, China; 5School of Physical Education, Shanghai University of Sport, Shanghai 200483, China; 6Centre for Active Living and Learning, University of Newcastle, Callaghan, NSW 2308, Australia; 7College of Human and Social Futures, University of Newcastle, Callaghan, NSW 2308, Australia

**Keywords:** physical fitness, motivation, enjoyment, well-being, physical education, China

## Abstract

Purpose: The purpose of this study is to investigate the relationship between physical fitness (PF) level and Chinese middle-school students’ perceived well-being, motivation, and enjoyment. Method: In this study, the participants were randomly selected from 16 Beijing-based middle schools in China. In addition to the collection of demographic data (i.e., gender, age, and parental education), tools including the Warwick-Edinburgh Mental Well-Being Scale, the Behavioural Regulation in Sport Questionnaire, and the Physical Activity Children’s Enjoyment Scale were employed for collecting data on other variables of well-being, motivation, and enjoyment, respectively. Results: A total of 1678 adolescents (M age = 14.66 years, SD = 1.80) participated in this study. According to the results, significant gender differences could be observed in the PF categories of mean age, BMI, vital capacity, 50-m sprint, muscular strength, and flexibility (all *p* < 0.05). In contrast, there was no significant gender difference in the levels of well-being, motivation, and enjoyment observed, with similar scores for boys and girls (*p* > 0.05). Indeed, adolescents with low BMI scores and high levels of vital capacity, muscular strength, and flexibility showed better well-being, motivation, and enjoyment. However, no association was observed between timed sit-ups and pull-ups and well-being, motivation, and enjoyment. Conclusions: This study produced preliminary findings on the relationship between PF and Chinese adolescents’ perceived well-being, motivation, and enjoyment. To improve the health of adolescents, it is necessary to adopt prospective and experimental research designs for advocating for fitness education in school-based programs in future research.

## 1. Introduction

Physical activity (PA) and physical fitness (PF) are significant health-related factors for the youth population [1,2]. PF, which can be achieved by PA, is the ability of the body to adapt to the surrounding environment [3,4,5]. Through the relative flexibility of PA, people can choose the most suitable form of exercise and level of exercise intensity according to their own needs in various circumstances to boost their PF and maintain their health [6]. Regular participation in PA, as well as promoting acceptable levels of PF, could not only improve mental health [7,8] and body and tissue functions [2,9,10] but also prevent and treat physical diseases [11,12,13]. Participating in aerobic exercise enhances cardiorespiratory fitness and is an effective treatment for several diseases, such as cardiovascular disease, obesity, hypercholesterolemia, and diabetes, by helping to reduce body fat [14,15,16].

According to Tomkinson et al. (2018) [17], PF consists of four components: the ability to perform PAs that require speed, endurance, strength, and flexibility. Thus, PF is a well-established marker of an individual’s health [18,19,20]. PF is categorized into health-related and skill-related fitness [21]. Health-related physical fitness (HRPF) consists of five components: cardiorespiratory fitness, body composition, muscular strength, muscular endurance, and flexibility [15,22]. A large body of research has demonstrated the importance and health benefits of higher levels of HRPF in children and adolescents [5,18,23,24].

The lifestyle habits and diet patterns of Chinese students have undergone major changes during the 21st century due to social and technological developments and changes in the public consciousness [25,26,27]. At present, assessments of Chinese students’ PF point to steadily increasing obesity rates in correlation with declining fitness levels [28]. However, these patterns tend to vary by gender and region. Assessments of students’ core strength, aerobic exercise capacity, and flexibility, as well as other indicators, have highlighted that physical fitness levels appear to be higher among female students than male students. Overall, 8% of Chinese students fail to meet the Chinese National Student Physical Fitness Standard (CNSPFS) [26].

Physical education (PE) at school involves group participation for children and adolescents [29,30]. Thus, it arguably facilitates positive PA experiences [31]. As PE programs in middle school—especially in early adolescence—can provide students with positive emotional experiences and motivation, this will be conducive to encouraging them to participate in PA in the future [32]. As demonstrated by prior studies, enjoyment is often a crucial factor in PA [33,34]. In this context, enjoyment is a multi-dimensional concept that includes excitement, capacity perception, and emotion, or more specifically, a positive effect related to liking, feelings of pleasure, and fun [35]. It has been highlighted by previous studies that enjoyment in PE is directly linked to the youth population’s engagement in PA in both PE [36] and in their spare time [37]. Hence, well-being can be defined as the experience of positive psychological functions [38]. Students’ well-being has been widely studied in the field of PE due to its importance [39]. One study on the perception of PE in middle school indicated that receiving support from teachers, satisfaction, and autonomous incentives are positively correlated with well-being [34]. On the other hand, PE has also been revealed to be negatively associated with ill-being [40].

In Western countries, extensive research has been conducted to demonstrate the relationship between physical fitness and perceived well-being, motivation, and enjoyment. Martínez-López et al. (2015) [41] surveyed 2293 Spanish students via a questionnaire and highlighted that low exercise frequency and a sedentary lifestyle (which translates into low fitness levels) are strongly associated with low well-being among Spanish teenagers. Similarly, Martínez-López et al. (2015) [42] studied 576 adolescents in the UK, measured their fitness levels, depressive symptoms, and quality of life, and found that cardiorespiratory fitness and BMI were linked directly and indirectly with mental well-being and quality of life. According to a study on the relationship between exercise and depression in European adults, there is a link between vigorous physical activity, lower levels of depression, and high enjoyment of life [43]. A similar study even found that physical activity can be used to treat mental health problems. On this basis, the necessity of encouraging vigorous physical activity and fitness is highlighted as a way to counteract the increasingly prevalent mental issues among teenagers [44]. However, only a small number of studies on this topic have thus far been conducted in a Chinese context. Ho et al. (2017) [45] found that a sports-based program had a positive effect on the physiological and mental well-being of adolescents in Hong Kong. As pointed out by Poon [46], 10 weeks of training in aerobic exercise can significantly improve cognitive functions and well-being (e.g., enjoyment) and reduce ill-being (e.g., negative affect) among adolescents.

The 40 years of economic reforms in China, the largest developing country, have had a significant impact on health and lifestyle, including an increased level of physical inactivity and BMI (unhealthy weight) and declined levels of physical fitness among adolescents [47,48]. While physical health is of vital importance in regard to health problems, there has also been a concerning rise in mental health issues among these groups that also require addressing. Garrido et al. (2019) [49] have argued that the prevalence of mental health issues, particularly depression, is increasing among youth in China. However, studies have rarely established a link between students’ physical fitness and self-reported well-being, motivation, and enjoyment. Due to insufficient sample sizes, these findings may apply to be validated in further studies with larger sample sizes to provide more substantial evidence in the Chinese population.

To fill this gap in the research, this study aimed to explore the relationship between students’ PF levels and their perceived well-being, motivation, and enjoyment in Chinese middle schools. It can be hypothesized that better physical fitness could be associated with high levels of well-being, motivation, and enjoyment among Chinese adolescents and demographic characteristics (e.g., grade, ethnicity, parent education, and BMI) affect physical fitness. The knowledge gained through this study may facilitate the development of health promotion policies and programs for Chinese adolescents.

## 2. Materials and Methods

### 2.1. Study Design and Participants

In this study, a cross-sectional survey was conducted in Beijing, China, from August to December 2021. According to the specific geographical, demographic, and socioeconomic levels of the districts, 16 middle schools were randomly selected from 16 administrative districts in Beijing by using a computer-based random number-producing algorithm to select. For every school, one, two, or three classes of each grade were chosen randomly by a contact who visited every school.

### 2.2. Procedure

Once a potential school expressed interest in this study, the corresponding author emailed, phoned, or Zoom called the representative(s) (e.g., the principal and administrative staff) to elaborate on the study’s requirements and process. The self-report scales were applied at the school by the staff, teacher staff, or headmaster of the school, and they helped to manage and fill out the questionnaire in a paper-based form. As a result, 1807 adolescents (aged from 12 to 16), who were recruited by the procedure below, constituted the original sampling results.

The participating adolescents were asked to provide data on the interested variables to be studied, while those who failed to offer the data needed in the study, such as covariates, outcomes, and independents, were eliminated from the initial samples. For the current study, 1678 respondents who had provided valid data for the variables to be studied were included. All the respondents participating in the current study and their guardians or parents were informed that it was completely voluntary to join the study, and informed consent was obtained from all subjects involved in the study. At the same time, the procedures followed the ethical standards of the latest version of the Helsinki Declaration for experiments [50], including human subjects. Additionally, the study protocol and procedure were approved by the Institutional Review Board (IRB) of the Shanghai University of Sport, and the grant number is 102772021RT071. The approval date was 24 May 2021.

### 2.3. Assessments

#### 2.3.1. Independent Variable (Physical Fitness)

All the assessments were examined by the trained assessors. The data were collected via the annual tests of physical fitness that were conducted during class hours, and the tests were compulsory, as provided by China’s Ministry of Education. In addition, the components of physical fitness were evaluated via the following tests with the revised 2014 version of the Chinese National Student Physical Fitness Standard (CNSPFS) battery [26], which was both valid and reliable in assessing the eight main components of physical fitness [51]. It is reliable and effective to use these test items to measure the physical fitness of Chinese teenagers. The test-retest reliability achieved for all the assessments conducted in the current study was an ICC (intra-class correlation coefficient) > 0.85, and the result was generally acceptable.

The surrogate assessment of body composition is BMI, which measures not only the height (cm) of the participants within 0.1 cm but also the weight (kg) within 0.1 kg through GMCS-IV, Jianmin, Beijing, China. During the anthropometric measurements, children were barefoot in light clothes. The BMI scores are obtained by the weight in kilograms divided by the squared height in meters (kg/m^2^). BMI = weight (kg)/height (m^2^). By following the procedure of the International Standards for Anthropometric Assessments (ISAK), two readings were recorded for every measurement, and a third reading was recorded if the difference was greater than 10% [52]. The final results were obtained by working out the average value of the readings.

#### 2.3.2. Lung Vital Capacity (VC)

In a quiet setting for assessment, spirometry was adopted to evaluate the vital capacity (VC) of the children [48]. By definition, VC refers to the maximum air volume (measured in milliliters) that a child expels from the lungs after taking a deep breath. The test was performed 3 times for every child, and the best performance of a child in the 3 tests was recorded (FHL-001, Jianmin, Beijing, China).

#### 2.3.3. 50 m Sprint Run

This sprint test was performed on a clear and flat ground where the respondents were asked to run straightforwardly for 50 (m) [48] through the use of the timing gates (LK-E3801, Jianmin, Beijing, China). Every respondent took one test. Namely, a single maximum sprint and their performances were taken down and corrected to 0.1 s. All the respondents were asked to take the test.

#### 2.3.4. Aerobic Fitness

Aerobic fitness was evaluated by organizing a long-distance race, namely 1000 m for boys and 800 m for girls [48]. Every participant stood at the starting line and needed to complete the 800 or 1000 m as fast as possible. The nearest 0.1 s was recorded as the running performance through the use of the timing gates (LK-E3801, Jianmin, Beijing, China).

#### 2.3.5. Muscular Strength

Muscular strength was checked by organizing a standing long jump, in which participants put their feet together behind a line on the ground [48]. Furthermore, the students leaped forward and tried to reach as far as they could, and the distance from the line they jumped from to the nearest contact point (the back of the heel) on the landing point was measured. Afterward, the score was determined as the distance from the start line to the heel of the closest foot, and each participant’s best score of the three attempts was retained (HJ-201, Jianmin, Beijing, China).

#### 2.3.6. Timed Sit-Ups (Girls)

To check students’ abdominal muscle endurance, they were also instructed to do sit-ups within a specific time [48]. Participants were instructed to do as many sit-ups within one minute. The sit-ups they did within one minute were counted by the testing staff. To do a standard sit-up, the student should lie down with their knees bent and put their feet flat on the mat, and put their hands behind their head with fingers crossed. The participants elevated their trunks until their elbows touched their thighs. Then, participants lowered their shoulder blades to the mat to return to the starting position. The number of completed repetitions was recorded as the final score.

#### 2.3.7. Pull-Ups (Boys)

Pull-ups were used to evaluate the muscular endurance of the upper part of the body [48]. The children made a long jump in an upright position, with their arms fully extended, and held an overhead bar with their overhand grip. Afterward, their arms were used to pull the body up until the chin was off the bar top, lowering their bodies again to the arm’s outstretched position. The number of completed repetitions was recorded as the final score.

#### 2.3.8. Flexibility

Based on the result of the sit-and-reach test, the flexibility of the lower part of the body was measured [48]. Under the instructions, the participants sat down, and their knees were fully extended. Further, their feet were firmly placed against vertical supports. Along a measuring line, they were requested to reach forward with their hands as far as possible. The best score of the two attempts was recorded. (Corrected to 0.1 cm), (WTS-600, Jianmin, Beijing, China).

#### 2.3.9. Outcome Variables (Well-Being, Motivation, Enjoyment)

Well-being was assessed using the Warwick-Edinburgh Mental Well-being Scale (WEMWBS), which consisted of 14 questions using a 5-point Likert Scale (ranging from 1 = “disagree a lot” to 5 = “agree a lot”, e.g., over the last 2 weeks, I have been feeling optimistic about the future) [53]. Possible scores range from 14 to 70, with a higher score indicating a higher well-being level. This scale and items have previously shown good convergent and divergent validity and reliability in Chinese children (Cronbach’s α = 0.889) [54].

Behavioral Regulation in Sport Questionnaire (BRSQ) was used to assess motivation [55,56], which consisted of 23 items using a 5-point Likert Scale (ranging from 1 = “disagree a lot” to 5 = “agree a lot”, e.g., I am in PE because the benefits are important to me). Possible scores range from 23 to 115, with a higher score indicating a higher motivation level; validated for use with Chinese children (χ^2^/df (3.2), CFI (0.88), and RMSEA (0.07)) [57].

Enjoyment of Sport was assessed using an adapted version of the Physical Activity Children’s Enjoyment Scale, using a 5-point Likert Scale (via 16 questions e.g., ranging from 1 = “disagree a lot” to 5 = “agree a lot”, e.g., When I am at sports training, I enjoy it) [58]. Possible scores range from 16 to 80, with a higher score indicating a higher enjoyment level, validated for use with Chinese children (Cronbach’s α = 0.91) [59].

### 2.4. Controlling Variables

Demographic information on study participants’ grades, gender, ethnicity, and parent’s education levels was measured by a self-reported questionnaire. These demographic factors were treated as covariates in further statistical analysis.

### 2.5. Statistical Analysis

After cleaning invalid and abnormal values from all independent and dependent variables (Figure 1), the final analytical sample size was 1678. All statistical analyses were performed using IBM SPSS software (Statistics 26, IBM Corporation, Chicago, IL, USA). The data were tested for normality with the Shapiro–Wilk test. First, the features of the samples to be studied were generalized through descriptive statistics. The independent t-tests had been performed to check if there was any difference between girls and boys concerning the evaluated variables. Continuous variables were expressed by the mean with standard deviation (mean ± standard deviation) in the case of continuous variables such as age, while the number (*n*) and percentage (%) were adopted to describe the categorical variables such as residence and grade. Secondly, linear regression was performed to analyze the relationship between age, ethnicity, gender, parent education, physical fitness (independent variables), and perceived well-being, motivation, and enjoyment (dependent variables). For all the models, the results had been adjusted and turned into a non-standard coefficient (β), and the confidence interval was 95% (95% CI). The statistical significance was set as *p* < 0.05.

## 3. Results

Table 1 shows the descriptive characteristics of the study participants based on gender. Overall, a total of 1678 adolescents participated in the final statistical analysis, of which 44.1% were boys and 55.9% were girls. According to the descriptive statistical analysis, the mean age, BMI, vital capacity, 50-m sprint, muscular strength, and flexibility were 14.4 years old, 22.2, 3737.0 mL, 195.3 cm, and 17.7 cm, respectively, indicating significant differences between boys and girls. The average 1000-m for boys and 800-m for girls were 3.5 min and 3.9 min, respectively. The average number of pull-ups for boys and timed sit-ups for girls were 5.3 and 44.0, respectively. However, no significant gender differences in well-being, motivation, and enjoyment attributes were observed, with similar scores in boys and girls.

Table 2 presents the relationships between physical fitness and perceived well-being, motivation, and enjoyment, as well as the demographic factors of adolescents that were analyzed based on multivariable generalized linear models. As shown in Table 2, the grade could significantly influence the perceived well-being (boys: β: −2.767; 95% CI: −3.776 to −1.757), motivation (boys: β: −6.234; 95% CI: −7.554 to −4.913), and enjoyment (boys: β: −0.965; 95% CI: −1.472 to −0.458, girls: β: −1.441; 95% CI: −2.740 to −0.143). By comparing participants on the demographic characteristic of parent education, it was found that parent education significantly affected perceived well-being in girls (β: 2.377; 95% CI: 1.385 to 3.360) but had no association with well-being or enjoyment in boys. Additionally, BMI and vital capacity significantly affected the three variables of well-being, motivation, and enjoyment for boys, respectively. Conversely, the 50-m sprint presented showed opposite trends for the three variables based on gender, with greater effects on well-being, motivation, and enjoyment in girls than in boys. Particularly, the 50-m sprint exerted a significant effect on girls’ motivation (β: −0.442; 95% CI: −0.617 to −0.267) and enjoyment (β: −0.224; 95% CI: −0.359 to −0.089). Beyond that, muscular strength was significantly negatively related to enjoyment in boys (β: −0.019; 95% CI: −0.036 to −0.001), and there was a great association between BMI and enjoyment in girls (β: 0.072; 95% CI: 0.004 to 0.139). In comparison to enjoyment, muscular strength was significantly associated with well-being (boys: β: 0.072; 95% CI: 0.037 to 0.108, girls: β: 0.098; 95% CI: 0.042 to 0.154) and motivation (boys: β: 0.147; 95% CI: 0.101 to 0.193, girls: β: 0.168; 95% CI: 0.139 to 0.196). Furthermore, flexibility presented a weaker relationship with well-being and enjoyment in boys but had a more significant effect on motivation in this gender group (β: 0.281; 95% CI: 0.210 to 0.353).

## 4. Discussion

This study was the first to examine the relationship between physical fitness and the perceived well-being, motivation, and enjoyment of adolescents from a Chinese background during physical education. This study found that the factors of BMI, vital capacity, muscular strength, and flexibility were individually and substantially associated with well-being, motivation, and enjoyment among adolescents. Adolescents with low levels of BMI and high levels of vital capacity, muscular strength, and flexibility experienced better well-being, motivation, and enjoyment. However, no association between timed sit-ups, pull-ups, well-being, motivation, or enjoyment was observed.

The findings of this study suggested that gender plays a crucial role in a student’s physical fitness, as significant differences between boys and girls were observed. To summarize, the boys performed better than the girls in terms of lung vital capacity, the 50-m sprint, and muscular strength. In comparison, the girls performed better than the boys on the flexibility test. Likewise, researchers worldwide have found that girls tend to demonstrate a lower sporting ability than boys, which is evident from an early age and continues to influence the physical aspects of their lives later on [60,61]. Regarding the mechanism of this gender difference, given the differences in muscle fiber structure and the female motor organs, it is reasonable to conclude that women tend to have more flexible motor organs than men [62]. It is noteworthy that no significant gender difference in terms of the students’ perceived well-being, motivation, or enjoyment was found in this study. Furthermore, the almost immaterial gender differences in the students’ subjective attitudes towards physical activities found by the study could be the result of a good class environment, positive dynamics [63], equal access to exercise equipment, and shifting sociocultural norms that have lessened the gender gap in attitudes towards physical activities [64].

One key factor that seems to have influenced the three outcome variables of well-being, motivation, and enjoyment among both genders was the participants’ school grade groups. According to a study conducted with Chinese middle-school students, adolescents in different grade groups demonstrated different levels of participation and enjoyment during physical activities, in which the two major factors were the influence of parents and financial independence [65,66]. While students may be psychologically more independent from their parents as they get older, they are also more likely to participate in more expensive sports, triggering physical health inequalities among different grade groups caused by socioeconomic disparities. The results of this study were in line with this finding.

Several researchers have discovered a connection between a student’s BMI and his or her level of physical fitness. In a study conducted with 70 middle-school students aged from 13 to 14 in Indonesia, researchers found that students with higher BMI scores had lower levels of physical fitness, and that an increase in a student’s fat-free mass could lead to a significantly higher level of physical fitness [67]. It was also demonstrated that the BMI evaluations could significantly affect both male and female students’ mental health. Notably, the BMI evaluations exerted a greater impact on the boys than on the girls in all three of the outcome variables of the study. This finding could be linked to the fact that China has seen a steady annual increase in BMI among its adolescents, with a higher average BMI score being recorded for boys [68,69]. While a student’s BMI can certainly be affected by several socioeconomic factors, this finding may nevertheless indicate to educators that mental well-being is closely associated with a person’s physical health. Notably, there was a significant association between BMI and enjoyment in girls, which was supported by previous studies [70,71]. This could be explained by a positive or enjoyable PE/sports learning environment, as hostile and vigorous PA in PE could reduce students’ passion and enjoyment.

Numerous studies have demonstrated the significance of physical fitness to one’s mental health, especially in adolescence. Beyond that, several relevant studies have highlighted a positive correlation between students’ cardiorespiratory fitness and a lower risk of depression, better muscular strength, and higher self-esteem, as well as flexibility and psychological improvement. These previous research findings, which found a significant correlation between students’ physical fitness and their well-being (r = 0.17, *p* = 0.021) [72], are in line with the results of this study. In addition to the aforementioned demographic factors, other important factors include lung vital capacity, muscular strength, and flexibility. Students with higher scores in these three areas also had higher levels of well-being, motivation, and enjoyment. Interestingly, despite the positive correlation between students’ perceived well-being and their lung vital capacity and muscular strength, there was little correlation between their perceived well-being and their flexibility. According to previous studies, boys with a higher vital capacity showed significantly higher levels of well-being and enjoyment during PA than girls [73,74]. Girls with a higher vital capacity demonstrated much higher motivation than boys, while boys tended to be more motivated and enjoyed physical activities more if they had better flexibility [75]. Muscular strength was also revealed to have a significant positive impact on both male and female students’ well-being and motivation. Our results indicated that a high level of muscular strength could result in a low level of enjoyment, which is inconsistent with the previous studies [76,77,78].

### 4.1. Study Strengths and Limitations

Firstly, the main strength of this study was that it used a relatively large sample size for the first time to examine the relationship between physical fitness and perceived well-being, motivation, and enjoyment in Chinese students during physical education. Secondly, the results of the comprehensive physical fitness test in this study, which was designed to evaluate flexibility, aerobic fitness, muscular strength, vital capacity, and body composition, were obtained from a nationwide standardized test and are thus, highly reliable. Thirdly, the participants were enrolled through random sampling, thereby enhancing the validity of the study.

However, there are several limitations that should be acknowledged. Firstly, the cross-sectional design of the study may not explain the causal relationship satisfactorily. Thus, prospective and experimental research designs should be adopted to better understand the causal relationship between adolescents, well-being, motivation, and enjoyment, to encourage a higher intervention success rate. Secondly, as the PF assessments were conducted in schools in an open testing environment rather than in a laboratory environment, this may have affected the accuracy of the results. The data collected by this study should be cautiously interpreted. Thirdly, as the samples were only collected from 16 middle schools in Beijing, they may not be fully representative of middle-school students across China. For this reason, the wider applicability of the research findings of this study is limited. Finally, there may have been some biases in the use of a questionnaire for data collection.

### 4.2. Practical Implications

The findings of this study have some practical implications for improving the physical and mental well-being of Chinese students. Firstly, equal access to physical education apparatus for both boys and girls will be conducive to lessening the gender gap in physical fitness. Thus, creating a holistic educational environment involving inclusive and mindful teaching techniques will help to improve the general outcomes of the physical activity for both genders. Secondly, it is arguable whether helping students to achieve academic success or even giving students more validation of their academic achievements can positively influence their mental health and well-being. Thirdly, this study also highlights that helping students to maintain their BMI within a healthy range will be favorable to significantly improving their mental health. Possible methods to achieve this may include adding nutrition classes to the curriculum, teaching students about mindful eating, and incorporating more vegetables and proteins into school meals. Indeed, any other practical recommendations aimed at improving either a student’s physical or mental well-being are supported by the findings of this study, as the two have been verified to be closely correlated.

## 5. Conclusions

To conclude, this study has provided new insight into the relationship between PF and perceived well-being, motivation, and enjoyment in Chinese adolescents, an issue that is rarely explored in academia in the public health field. Overall, this study has stressed the role of physical fitness in the well-being, motivation, and enjoyment of adolescents. Thus, it is essential to investigate whether the findings of this study can be reproduced in further studies of other population groups such as children, university students, and people from low- and middle-income countries. Apart from that, prospective and experimental research designs should be taken into account to better understand the causal association between PF and mental well-being.

## Figures and Tables

**Figure 1 children-10-00111-f001:**
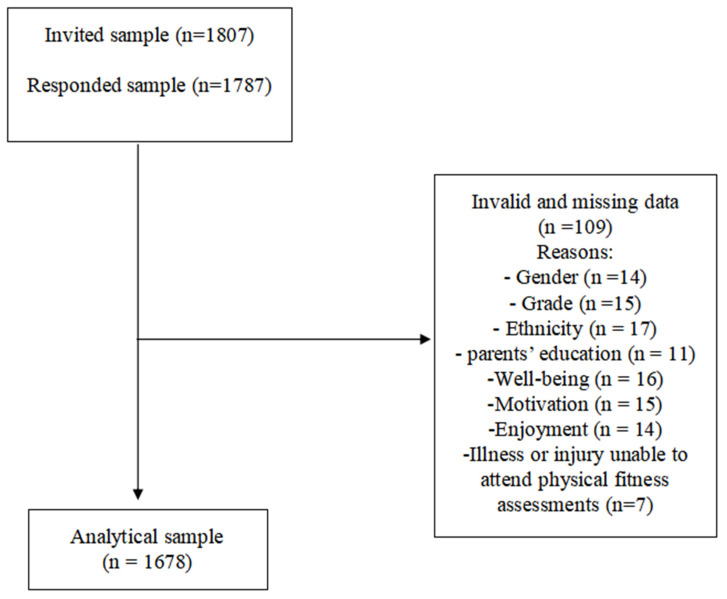
The procedure used for cleaning invalid and missing data in this study.

**Table 1 children-10-00111-t001:** Descriptive Characteristics of the Study Sample.

Variables	Overall (*n* = 1678)	Boys (*n* = 769)	Girls (*n* = 909)	*p*-Value
	Mean ± SD	Mean ± SD	Mean ± SD	
Age (years)	14.66 ± 1.80	14.43 ±1.7	14.89 ± 1.9	0.00 ******
BMI	22.28 ± 3.8	23.4 ± 4.0	20.8 ± 3.9	0.00 ******
Physical Fitness				
Vital capacity (mL)	3737.04 ± 994.2	4390.3 ± 795.8	2913.0 ± 467.4	0.00 ******
50-m sprint (sec)	8.0 ± 0.9	7.36 ± 0.5	8.8 ± 0.5	0.00 ******
1000-m run (min)		3.5 ± 0.3		
800-m run (min)			3.9 ± 0.3	
Muscular strength (cm)	195.3 ± 32.3	217.1 ± 23.2	167.9 ± 18.6	0.00 ******
Pull-ups (boys) (reps)		5.3 ± 6.4		
Timed sit-ups (girls) (reps)			44.0 ± 10.6	
Flexibility (cm)	17.7 ± 6.4	16.7 ± 6.8	19.0 ± 5.6	0.00 ******
Well-being	45.8 ± 13.2	45.5 ± 13.2	46.2 ± 13.1	0.52
Motivation	71.5 ± 17.0	70.9 ± 17.0	72.2 ± 16.9	0.35
Enjoyment	40.8 ± 10.4	40.1 ± 10.5	41.6 ± 10.2	0.09

Data were described as *n* (%) or mean ± SD; BMI, Body Mass Index; ** Significant difference between boys and girls, *p* < 0.01 (two-tailed).

**Table 2 children-10-00111-t002:** Multivariable Generalized Linear Models Evaluating the Associations Between the Physical Fitness and Perceived Well-being, Motivation and Enjoyment.

Variables	Demographic Characteristics	Boys (*n* = 798) β (95% CI)	Girls (*n* = 1009) β (95% CI)
Well-being	Grade	−2.767 (−3.776 to −1.757) **	0.425 (−0.616 to 1.466)
	Ethnicity	−0.877 (−3.788 to 2.035)	−1.095 (−3.564 to 1.374)
	Parent Education	0.697 (−0.695 to 2.088)	2.377 (1.385 to 3.360) **
	BMI	−0.131 (−0.191 to −0.071) **	−0.102 (−0.148 to 0.157) **
	Vital capacity	0.066 (0.031 to 0.100) **	0.003 (−0.053 to 0.060)
	50-m sprint	−0.019 (−0.105 to 0.068)	−0.069 (−0.039 to 0.177)
	1000-m run	−0.013 (−0.047 to 0.072)	
	800-m run		−0.086 (−0.117 to 0.155)
	Muscular strength	0.072 (0.037 to 0.108) **	0.098 (0.042 to 0.154) **
	Pull-ups (boys)	0.24 (0.001 to 0.047) *	
	Timed sit-ups (girls)		0.034 (−0.020 to 0.088)
	Flexibility (cm)	0.020 (−0.035 to 0.074)	0.196 (0.125 to 0.267)
Motivation	Grade	−6.234 (−7.554 to −4.913) **	0.437 (−1.250 to 2.214)
	Ethnicity	1.329 (−2.482 to 5.140)	−2.320 (−6.322 to 1.682)
	Parent Education	1.635 (−0.186 to 3.457)	−1.516 (−3.123 to 0.091)
	BMI	−0.083 (−0.161 to −0.005) *	−0.056 (−0.145 to 0.032)
	Vital capacity	0.027 (−0.019 to 0.072)	0.188 (0.097 to 0.280) **
	50-m sprint	−0.068 (−0.182 to 0.045)	−0.442 (−0.617 to −0.267) **
	1000-m run	−0.021 (−0.099 to 0.057)	
	800-m run		−0.417 (−0.529 to −0.306) **
	Muscular strength	0.147 (0.101 to 0.193) **	0.168 (0.139 to 0.196) **
	Pull-ups (boys)	−0.018 (−0.048 to 0.012)	
	Timed sit-ups (girls)		0.046 (−0.042 to 0.134)
	Flexibility (cm)	0.281 (0.210 to 0.353) **	0.048 (−0.066 to 0.162)
Enjoyment	Grade	−0.965 (−1.472 to −0.458) **	−1.441 (−2.740 to −0.143) **
	Ethnicity	2.448 (0.986 to 3.911) *	1.225 (−1.855 to 4.305)
	Parent Education	0.402 (−0.297 to 1.101)	1.132 (−0.105 to 2.369)
	BMI	0.028 (−0.002 to 0.057)	0.072 (0.004 to 0.139) *
	Vital capacity	0.045 (0.028 to 0.063) **	0.021 (−0.092 to 0.050)
	50-m sprint	−0.008 (−0.051 to 0.036)	−0.224 (−0.359 to −0.089) **
	1000-m run	−0.015 (−0.025 to 0.044)	
	800-m run		−0.063 (−0.149 to 0.023)
	Muscular strength	−0.019 (−0.036 to −0.001) *	−0.032 (−0.087 to 0.023)
	Pull-ups (boys)	−0.007 (−0.018 to 0.004)	
	Timed sit-ups (girls)		0.151 (0.083 to 0.218) **
	Flexibility (cm)	0.044 (0.016 to 0.071) **	0.042 (−0.046 to 0.130)

Data are presented as β coefficient (95% CI). * 0.05, ** 0.01.

## Data Availability

The original contributions presented in this study are included in the article. Further inquiries can be directed to the corresponding author.

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
