# Peer review of "The Relationship between Physical Fitness and Perceived Well-Being, Motivation, and Enjoyment in Chinese Adolescents during Physical Education: A Preliminary Cross-Sectional Study"

_children, 2023, doi:10.3390/children10010111_

Round 1

Reviewer 1 Report

Reviewer Comment

The subject of the research is well grounded and well planned. The writing of the manuscript is generally suitable for the article writing template. The studies and the entire text are nicely explained with reference sources. The methodological nature of the study is well justified. Along with these, there are some parts of the manuscript that need to be revised. Please make these arrangements carefully.

Revision

1.      p-values > 0.05It is sufficient to write the p expression here alone. Express as p> 0.05

2.      Page 2-3, line 53, 69, 5, 91, 104 In many parts of the introductory part, an example sentence is used by using the phrase "for example". This language you are using is not suitable for article writing. Delete this word from the entire introduction.

3.      “Due to insufficient sample sizes, these findings need to be validated in further studies with larger sample sizes to provide more substantial evidence in the Chinese population.” In this sentence, an expression meaning necessity is used. It disrupts the flow of the paragraph. Edit this sentence by removing the requirement statement.

4.      Therefore, it is necessary to conduct this research in the Chinese context to investigate how PF and students' perceived well-being, motivation, and enjoyment mutually affect each other, and thus provide the groundwork for future analysis and education practitioners.” In this sentence at the end of the introduction, it is wrong to use a meaning of necessity. It disrupts the flow of the sentence. It cannot be stated as a requirement that you want to implement this study in China. Edit this sentence so that it is an objective sentence rather than a necessity.

5.      Add the hypothesis of the research at the end of the introduction.

6.      “The surrogate assessment of body composition is BMI, which measures not only the height (cm) of the participants within 0.1 cm but also the weight (kg) within 0.1 kgs through GMCS-IV, Jianmin, Beijing, China. During the anthropometric measurements, children were barefoot in light clothes. The BMI scores are obtained by the weight in kilograms divided by the squared height in meters (kg/m2). BMI= weight (kg)/height (m2). By following the procedure of the International Standards for Anthropometric Assessments (ISAK), two readings were recorded for every measurement, and third reading was recorded if the difference is greater than 10%. The final results were acquired by working out the average value of readings.” Indicate the source for the quotations.

7.      Page 4-5, line 156-203; All descriptions must include a source reference.

8.      Is there a cut-off value in the total scores in the questionnaires used? Please add to the explanations what does the increase in the score mean.

9.      “All statistical analyses were performed using IBM SPSS software (Statistics 26, IBM Corporation, Chicago, USA)” Add your source.

10.  In general, although the sources above 2005 can be accepted, provided that they are few, your 34th source is a very old source. It is very important and essential that the sources are up-to-date in the writing of an article. Therefore, revise this resource.

Author Response

Responses to Reviewer 1 Comments

Dear reviewer,

Thank you for your time and valuable comments. We have provided a point-by-point response to each of your comments and suggestions and have made the appropriate changes to the manuscript. We believe the paper has improved significantly because of the review process. Finally, merry Christmas and Happy New Year to you and your family. May this time of the year be truly blissful and enjoyable for all of us.

Response to Reviewer comments

  1. “p-values > 0.05” It is sufficient to write the p expression here alone. Express as p> 0.05

Response: Thank you for suggestion, we have revised as the suggestion, please see the 31 and 33.

  1. Page 2-3, line 53, 69, 5, 91, 104 In many parts of the introductory part, an example sentence is used by using the phrase "for example". This language you are using is not suitable for article writing. Delete this word from the entire introduction.

Response: Thank you for suggestion, we have removed the “for example” from the introduction.

  1. “Due to insufficient sample sizes, these findings need to be validated in further studies with larger sample sizes to provide more substantial evidence in the Chinese population.” In this sentence, an expression meaning necessity is used. It disrupts the flow of the paragraph. Edit this sentence by removing the requirement statement.

Response: Thank you for suggestion, please see the line 113-115.

  1. “Therefore, it is necessary to conduct this research in the Chinese context to investigate how PF and students' perceived well-being, motivation, and enjoyment mutually affect each other, and thus provide the groundwork for future analysis and education practitioners.” In this sentence at the end of the introduction, it is wrong to use a meaning of necessity. It disrupts the flow of the sentence. It cannot be stated as a requirement that you want to implement this study in China. Edit this sentence so that it is an objective sentence rather than a necessity.

Response: Thank you for suggestion, we have removed this sentence.

  1. Add the hypothesis of the research at the end of the introduction.

Response: Thank you for suggestion, please see the line 118-122.

  1. “The surrogate assessment of body composition is BMI, which measures not only the height (cm) of the participants within 0.1 cm but also the weight (kg) within 0.1 kgs through GMCS-IV, Jianmin, Beijing, China. During the anthropometric measurements, children were barefoot in light clothes. The BMI scores are obtained by the weight in kilograms divided by the squared height in meters (kg/m2). BMI= weight (kg)/height (m2). By following the procedure of the International Standards for Anthropometric Assessments (ISAK), two readings were recorded for every measurement, and third reading was recorded if the difference is greater than 10%. The final results were acquired by working out the average value of readings.” Indicate the source for the quotations.

Response: Thank you for suggestion, we have updated the reference, please see line 165.

  1. Page 4-5, line 156-203; All descriptions must include a source reference.

Response: Thank you for suggestion, we have added all the references for these descriptions. Please see the line 169, 175, 181, 187, 196, 205, 213.

  1. Is there a cut-off value in the total scores in the questionnaires used? Please add to the explanations what does the increase in the score mean.

Response: Thank you for suggestion, please see the line 220, 228, and 233.

  1. “All statistical analyses were performed using IBM SPSS software (Statistics 26, IBM Corporation, Chicago, USA)” Add your source.

Response: Thank you for suggestion, however, IBM SPSS as a statistic tool which has been performed in our study.

  1. In general, although the sources above 2005 can be accepted, provided that they are few, your 34th source is a very old source. It is very important and essential that the sources are up-to-date in the writing of an article. Therefore, revise this resource.

Response: Thank you for suggestion, we have updated the reference.

Reviewer 2 Report

First of all I would like to congratulate the authors for their clear and precise work. However, the novelty is hardly visible, as well as the justification of this study in relation to those that already exist. In addition I have a few major comments: 

The abstract is too long and exceeds the recommended number of words. The method can be considerably reduced, it is not necessary to explain the statistical analysis. 

Why this study is important in China, there are already many studies on this topic, the authors should justify why the study is important and that the Chinese context differs from the rest. 

It is recommended that the authors divide the section "Study design and participants" into "design", "procedure", "participants" and "Ethics". 

In addition, the procedure should include how the schools were contacted, how the information was sent to the parents, whether or not they filled in a permission form, how the process of agreeing to participate in the study was for each student, etc. How the survey was carried out, online, in person, within or outside school hours.

In participants, specify the inclusion criteria. Also, those eliminated because they were eliminated and how many for each reason. 

Although the tests correspond to the Chinese National Student Physical Fitness Standard (CNSPFS), it is recommended that the authors include the reliability of each test specifically.

In the section "variable outcome", it is recommended that you specify what the total possible score is and what one or the other score means, i.e. a higher score on the questionnaire means higher or lower motivation, for example.

In the statistical analysis there is repeated t-test information. Specify which were taken as dependent and independent variables in the regression. 

If valid data were only obtained from 1,678 (line 129), why are data from 1807 (table 1) included?

Table 2 is poorly structured and not easy to read, "R2" and "F" are out of place.

In line 281 the authors mention that it is during PE, but how do they take this into account, on what basis do they make this specification. Moreover, this is not included in their objective. 

Author Response

Responses to Reviewer 2 Comments

Dear reviewer,

Thank you for your time and valuable comments. We have provided a point-by-point response to each of your comments and suggestions and have made the appropriate changes to the manuscript. We believe the paper has improved significantly because of the review process. Finally, merry Christmas and Happy New Year to you and your family. May this time of the year be truly blissful and enjoyable for all of us.

Response to Reviewer comments

First of all I would like to congratulate the authors for their clear and precise work. However, the novelty is hardly visible, as well as the justification of this study in relation to those that already exist. In addition I have a few major comments:

The abstract is too long and exceeds the recommended number of words. The method can be considerably reduced, it is not necessary to explain the statistical analysis.

Response: Thank you for suggestion, we have removed the statistical analysis from the abstract.

Why this study is important in China, there are already many studies on this topic, the authors should justify why the study is important and that the Chinese context differs from the rest.

Response: Thank you for the suggestion, please see the line 105-115.

It is recommended that the authors divide the section "Study design and participants" into "design", "procedure", "participants" and "Ethics".

Response: Thank you for the suggestion, please see the line, and the ethic can be found the line 129, 146 and 414, according to current journal requirements.

In addition, the procedure should include how the schools were contacted, how the information was sent to the parents, whether or not they filled in a permission form, how the process of agreeing to participate in the study was for each student, etc. How the survey was carried out, online, in person, within or outside school hours.

Response: Thank you for the suggestion, please see the line 130-135.

In participants, specify the inclusion criteria. Also, those eliminated because they were eliminated and how many for each reason.

Response: Thank you for the suggestion, please see the figure 1.

Although the tests correspond to the Chinese National Student Physical Fitness Standard (CNSPFS), it is recommended that the authors include the reliability of each test specifically.

Response: Thank you for the suggestion, however, the previous studies only reported the reliability of CNSPFS, rather than every item’s reliability. Please see the following publications:

  1. Yi, X., Fu, Y., Burns, R. D., Bai, Y., & Zhang, P. (2019). Body mass index and physical fitness among Chinese adolescents from Shandong Province: a cross-sectional study. BMC public health, 19(1), 1-10.
  2. Dun, Y., Ripley-Gonzalez, J. W., Zhou, N., Li, Q., Chen, M., Hu, Z., ... & Liu, S. (2021). The association between prior physical fitness and depression in young adults during the COVID-19 pandemic—a cross-sectional, retrospective study. PeerJ, 9, e11091.

In the section "variable outcome", it is recommended that you specify what the total possible score is and what one or the other score means, i.e. a higher score on the questionnaire means higher or lower motivation, for example.

Response: Thank you for the suggestion, please see the line 219-223, 228, and 233.

In the statistical analysis there is repeated t-test information. Specify which were taken as dependent and independent variables in the regression.

Response: Thank you for the suggestion, we have removed the repeated part and added dependent and independent variables in the regression, please see the line 250-251.

If valid data were only obtained from 1,678 (line 129), why are data from 1807 (table 1) included?

Response: Thank you for the correction, we really sorry to put the wrong number in the table1. We have corrected it, please see the table1.

Table 2 is poorly structured and not easy to read, "R2" and "F" are out of place.

Response: Thank you for the suggestion, we have removed the "R2" and "F", please see the table 2.

In line 281 the authors mention that it is during PE, but how do they take this into account, on what basis do they make this specification. Moreover, this is not included in their objective

Response: Thank you for the suggestion, all the physical fitness tests were conducted in the Physical Education classes, at the end of the regular academic semester.

Round 2

Reviewer 2 Report

All comments have been addressed. 

It is recommended to review the figure and put the N in the same format, all with or without ",".

The references are not in the format of the journal. 

Author Response

Thank you for your time and valuable comments.
